# Pregnancy complications and maternal birth outcomes in women with intellectual and developmental disabilities in Wisconsin Medicaid

Eric Rubenstein[1,2]*, Deborah B. Ehrenthal[3,4], David C. Mallinson[3,5], Lauren Bishop[1,6], Hsiang-Huo Kuo[4], Maureen Durkin[1,3]

1 Waisman Center, University of Wisconsin Madison, Madison, WI, United States of America, 2 Department of Epidemiology, Boston University School of Public Health, Boston, MA, United States of America, 3 Department of Population Health Sciences, University of Wisconsin School of Medicine and Public Health, Madison, WI, United States of America, 4 Department of Obstetrics and Gynecology, University of Wisconsin School of Medicine and Public Health, Madison, WI, United States of America, 5 Center for Demography and Ecology, University of Wisconsin-Madison, Madison, WI, United States of America, 6 Sandra Rosesnbaum School of Social Work, University of Wisconsin-Madison, Madison, WI, United States of America

* erubens@bu.edu

**Data Availability Statement:** Data cannot be shared publicly because of their proprietary ownership by the Wisconsin Department of Health

## Abstract

### Background

Women with intellectual and developmental disabilities (IDD) may face greater risk for poor pregnancy outcomes. Our objective was to examine risk of maternal pregnancy complications and birth outcomes in women with IDD compared to women without IDD in Wisconsin Medicaid, from 2007–2016.

### Methods

Data were from the Big Data for Little Kids project, a data linkage that creates an administrative data based cohort of mothers and children in Wisconsin. Women with ≥1 IDD claim the year before delivery were classified as having IDD. Common pregnancy complications and maternal birth outcomes were identified from the birth record. We calculated risk ratios (RR) using log-linear regression clustered by mother. We examined outcomes grouped by IDD-type and explored interaction by race.

### Results

Of 177,691 women with live births, 1,032 (0.58%) had an IDD claim. Of 274,865 deliveries, 1,757 were to mothers with IDD (0.64%). Women with IDD were at greater risk for gestational diabetes (RR: 1.28, 95% CI: 1.0, 1.6), gestational hypertension (RR: 1.22, 95% CI: 1.0, 1.5), and caesarean delivery (RR 1.32, 95% CI: 1.2, 1.4) compared to other women. Adjustment for demographic covariates did not change estimates. Women with intellectual disability were at highest risk of gestational hypertension. Black women with IDD were at higher risk of gestational hypertension than expected under a multiplicative model.

and the Wisconsin Institute for Research on
Poverty. The data use agreement between the
author's and the agencies prohibit sharing of data
files. Wisconsin administrative data access can be
requested from the Institute for Research on
Poverty (contact via irp.wisc.edu) for researchers
with approved proposals who meet the criteria for
access to confidential data. Data were accessed for
this project through these means.

**Funding:** This work was supported by the Eunice
Kennedy Shriver National Institute for Child Health
and Human Development (R03HD099619- ER; T32
HD007014-42- DM; https://connections.
spectrumnews.org/) and the University of
Wisconsin-Madison Clinical and Translational
Science Award programme through the National
Institutes of Health National Center for Advancing
Translational Sciences (UL1TR00427-LB, KL2
TR002374-LB; https://ictr.wisc.edu/), by the
University of Wisconsin-Madison School of
Medicine and Public Health's Wisconsin
Partnership Program (DE; https://www.med.wisc.
edu/wisconsin-partnership-program/), and by the
University of Wisconsin- Madison Institute for
Research on Poverty (DE; https://www.irp.wisc.
edu/). This study was supported in part by a core
grant to the Waisman Center from the Eunice
Kennedy Shriver National Institute of Child Health
and Human Development (U54 HD090256- ER, LB,
DM; nichd.nih.gov). The funders had no role in
study design, data collection and analysis, decision
to public, or preparation of the manuscript.

**Competing interests:** The authors have declared
that no competing interests exist.

## Conclusions

Women with IDD have increased risk of pregnancy complications and adverse outcomes in Wisconsin Medicaid. Results were robust to adjustment. Unique patterns by IDD types and Black race warrant further exploration.

## Introduction

Intellectual and developmental disabilities (IDD)—which present before 18 years of age and are defined by limitations in adaptive, cognitive, and social functioning [1]—affect approximately 2.1 million adults [2]. As larger cohorts of children with IDD age into adulthood [3], health and service needs will change from mainly pediatric services to specialized adult care [4]. Clinicians and health care systems will face new challenges in serving a population at risk for disparities in health outcomes.

A specific area illustrative of health disparities in IDD is poor perinatal outcomes for women with IDD. Like women without IDD, women with IDD have the right to achieve their sexual health goals, including the choice to engage in sexual relationships and the choice to become a parent. However, historically and in recent decades women with IDD have been denied reproductive rights through forced sterilization, institutionalization, and stigma [5, 6]. Although attempts have been made to repudiate past injustices [5], individuals with IDD face discrimination and healthcare inequity that place them at greater risk for adverse pregnancy outcomes [7, 8]. The stigma surrounding pregnancy and the lack of adequate reproductive and sexual healthcare [9, 10] makes an at-risk population of women with IDD especially vulnerable during pregnancy.

A recent meta-analysis identified 10 studies of pregnancy outcomes for women with IDD suggesting that women with IDD are more likely than peers to experience complications [11]. Yet, studies of pregnancy in women with IDD are often confounded by socioeconomic status because a full population comparison groups does not mirror the lower socioeconomic profile of the IDD population [11]. Medicaid, a federal-state partnership to provide low- or no-cost health care, nursing home care, and personal care services to low-income individuals in the US, is a source of data that may allow for assessment of more like IDD and non-IDD groups. Further, little is known about pregnancy outcomes among specific IDD types (e.g. cerebral palsy, intellectual disability) and more work is needed, as there may be different associations with outcomes based on phenotypic and genetic presentation of the IDD [12]. Adults of color with disabilities often face compounded disparity that leads to excess poor health [13, 14], but literature is lacking surrounding pregnancy in women of color with IDD [15].

Our objective was to replicate and expand upon past findings focused on pregnancy complications and maternal birth outcomes in women with IDD. We used linked administrative data from the Big Data for Little Kids (BD4LK) project from 2007–2016 to characterize and compare pregnancy complications and maternal birth outcomes in 1,032 women with IDD and 176,665 women without IDD. Our focus is on relatively common complications that we can assess in birth records. We explored differences by IDD type and race. We hypothesized that women with IDD would be at higher risk for gestational hypertension, gestational diabetes, and caesarean delivery compared to women without IDD, with the highest risk for complications in women with intellectual disability. Additionally, we hypothesized that Black women with IDD would have higher risk of gestational diabetes and gestational hypertension compared to the expected independent risks associated with being Black and having IDD.

## Materials and methods

### Big Data for Little Kids

We used data from BD4LK, a longitudinal cohort of all Wisconsin birth records for in-state resident deliveries resulting in a live birth from 2007–2016 linked to administrative data sources, including Wisconsin Medicaid claims and encounters (hereafter 'claims'). BD4LK uses deterministic matching to link birth records to a Medicaid demographic base file using a mother's full name and birth date. BD4LK then links the file to claims using the Medicaid-specific person identifying number. In BD4LK, Medicaid claims from one year pre-delivery through delivery are available. BD4LK spans two versions of the birth record: the 1989 (2007–2010 records) and 2003 (2011–2016 records) Revision of the US Standard Certificate of the Live Birth. Shared variables were harmonized, although variables unique to the 2003 Revision are only available on 2011–2016 records. Further BD4LK methods can be found elsewhere [16–18].

Our sample consisted of all live births to women who had at least one Medicaid-paid delivery in Wisconsin during 2007–2016. Among 666,375 unique birth records in BD4LK, we linked 284,496 deliveries (42.6%) to women with Medicaid claims. We excluded records with imperfect linkages (e.g. missing maternal or child identifier; N = 1,825: <0.3%) and records of children born to women whose only observed Medicaid-paid delivery occurred in 2013, as data from Medicaid were unavailable for 2013 (N = 7,806: 1.2%). This yielded a final analytic sample of 274,865 birth records (41.2% of all BD4LK birth records) among 177,697 women.

BD4LK's first wave pulled claims that were bounded by 2007–2012, and the second wave pulled claims bounded by 2013–2016. As such, BD4LK does not have access to a full year of predelivery claims for women delivering in 2007, 2013, and early 2014. We generated a subsample excluding those years (N = 195,691: 71.2% of full analytic sample) for sensitivity analyses. Since we included all births to women who had any one delivery covered by Medicaid, a woman could have had a non-Medicaid covered delivery. Therefore, we ran a sensitivity analysis restricting to Medicaid covered deliveries only (N = 254,957: 92.8% of full analytic sample).

### Identifying IDD

We assessed maternal IDD using an algorithm that examined all International Classification of Disease 9 and 10 codes in available Medicaid claims. Women had differing amounts of time enrolled in Medicaid, but based on consistency in IDD claims over time [3] and ability to identify IDD during delivery-related hospitalizations [19], we believe the data were adequate for assessing IDD. Women with IDD qualify for Medicaid either by receiving a Social Security Disability Determination and/or by meeting income and asset requirements. IDD codes were identified from previous literature [20–22] (S1 Table). Women had to have one claim of IDD to be in the IDD group; we ran a sensitivity analysis using two claims and two claims on two different days. We grouped conditions into IDD-types: intellectual disability, genetic or chromosomal anomalies ('genetic conditions'), autism spectrum disorder ("autism"), and cerebral palsy. Women could have more than one IDD-type.

### Pregnancy complications and maternal birth outcomes

Data for maternal pregnancy complications and maternal birth outcomes were identified a priori from the birth record, specifically gestational diabetes, gestational hypertension, and common delivery complications (induced labor, precipitous labor [birth after <3 hours of regular contractions], prolonged labor [>20 hours for first time mother, >14 hours otherwise], caesarean-section, and maternal hospital transfer). We examined receipt of health services: timing of

entry into prenatal care, use of Special Supplemental Nutrition Program for Women Infants and Children during pregnancy (WIC; 2011–2016 birth records only- N = 157226). We assessed pre-pregnancy BMI (2011–2016 birth records only—N = 157,226) as a potential risk factor.

## Demographic variables

Maternal demographic characteristics at delivery were available from the birth record. We examined maternal race (white, Black, Asian, other), ethnicity (Hispanic, non-Hispanic), marital status (married, single), education (< high school, some high school, completed high school, ≥ some college), nativity, geographic classification of the birth county (based on National Center for Health Statistics 2013 Urban Rural Classification Scheme [23]), number of live births during the ten-year period (1, 2, ≥3), and total parity (first, second, third, or fourth or subsequent delivery). We categorized maternal age at childbirth in years (≤18, 19–24, 25–34, 35–39, ≥40). For women with multiple observed deliveries in the 10-year period, we present demographic characteristics from a randomly selected birth.

## Statistical analyses

We calculated descriptive statistics for women with and without IDD and by IDD-type. All analyses were restricted to live births. For each outcome we ran a multi-level log-linear regression calculating unadjusted risk ratios (uRR) and 95% confidence intervals (95% CI) comparing women with and without IDD and then by IDD-types. Outcomes with multiple levels (i.e. BMI) were dichotomized. We clustered by mother and used an exchangeable correlation structure to account for women having more than one birth. For prevalent outcomes (>5%) we calculated adjust risk ratios (aRR) models by adding covariates for maternal race, ethnicity, age, nativity, parity, and geographic classification of birth county to the previously described model. Lastly, we assessed race by IDD interaction on the risk ratio scale by adding a race term (white or Black, irrespective of ethnicity) and a race by IDD interaction term to the unadjusted models. Statistical significance was set at an alpha level of 0.1 for interaction analyses. We ran analyses using SAS version 9.4 (SAS Institute, Cary, NC). The data received for this secondary analysis were anonymized. The BD4LK project received a waiver of informed consent. This study was approved by the University of Wisconsin-Madison Institutional Review Board.

# Results

Of 177,691 women in our full sample, 1,032 (0.58%) had a Medicaid claim for IDD; those women delivered 1,757 children (0.64% of all births). Of the 1,032 women with IDD, 435 had claims for a genetic condition, 330 had claims for intellectual disability, 100 had claims for autism, 170 had claims for cerebral palsy, and 31 had claims for an 'other' IDD.

## Demographic characteristics

Our sample was predominantly of white race (Table 1). Approximately 25% of women with IDD reported some college or greater compared to 40% of women without IDD. The percentage of women with IDD who were foreign born (4.9%) or Hispanic (10.6%) was less than women without IDD (10.1% foreign born and 14.5% Hispanic). Categorized, mean, and median age at birth were similar for those with and without IDD.

By IDD-type (S2 Table), 33.8% of women with intellectual disability were Black and the percent of Black women in the other IDD-types was between 13% and 17%. Educational attainment was lower among women with intellectual disability (6.5% had ≥ some college

**Table 1. Demographic characteristics of women with a live Medicaid covered birth in Wisconsin, 2007–2016; by intellectual and developmental disability status.**

| | Women with intellectual and developmental disability | | Women without intellectual and developmental disabilities | |
| --- | --- | --- | --- | --- |
| | N = 1032 | | N = 176665 | |
| | N | % | N | % |
| **Maternal race** | | | | |
| White | 739 | 71.6 | 129724 | 73.4 |
| Black | 228 | 22.1 | 32558 | 18.5 |
| Asian | 44 | 4.3 | 10279 | 5.8 |
| Other | 21 | 2.0 | 4098 | 2.3 |
| **Hispanic ethnicity** | | | | |
| Hispanic | 109 | 10.6 | 25663 | 14.5 |
| Non-Hispanic | 923 | 89.4 | 150996 | 85.5 |
| **Foreign born** | | | | |
| Yes | 51 | 4.9 | 17823 | 10.1 |
| No | 981 | 95.1 | 158832 | 89.9 |
| **Maternal education** | | | | |
| <High school | 268 | 26.3 | 36820 | 21.0 |
| Completed high school | 496 | 48.6 | 71609 | 40.8 |
| Some college | 206 | 20.2 | 53246 | 30.3 |
| ≥ Completed college | 50 | 4.9 | 13833 | 7.9 |
| Missing | 12 | - | 1151 | |
| **Marital status** | | | | |
| Married | 284 | 27.5 | 57950 | 32.8 |
| Not Married | 748 | 72.5 | 118704 | 67.2 |
| Missing | | | 19 | |
| **Number of births, 2007–2016** | | | | |
| 1 | 560 | 54.3 | 106647 | 60.3 |
| 2 | 293 | 28.4 | 50113 | 28.4 |
| 3+ | 179 | 17.3 | 19899 | 11.3 |
| | **By births** | | | |
| | N = 1757 | | N = 273108 | |
| | N | % | N | % |
| **Maternal age at childbirth** | | | | |
| ≤18 | 68 | 3.8 | 8919 | 3.3 |
| 19–24 | 772 | 43.9 | 118667 | 43.5 |
| 25–29 | 486 | 27.7 | 81042 | 29.7 |
| 30–34 | 264 | 15.0 | 43932 | 16.1 |
| 35–39 | 124 | 7.0 | 16847 | 6.1 |
| ≥40 | 44 | 2.5 | 3691 | 1.4 |
| Mean age, SD | 26.5 | 6.0 | 26.3 | 5.7 |
| Median age, IQR | 26.0 | 8.3 | 25.6 | 8.1 |
| **Year of birth** | | | | |
| 2007 | 180 | 10.2 | 28375 | 10.4 |
| 2008 | 243 | 13.8 | 29032 | 10.6 |
| 2009 | 222 | 12.6 | 30153 | 11.0 |
| 2010 | 192 | 10.9 | 29242 | 10.7 |
| 2011 | 195 | 11.1 | 28983 | 10.6 |

*(Continued)*

**Table 1.**  (Continued)

| | Women with intellectual and developmental disability | | Women without intellectual and developmental disabilities | |
|---|---|---|---|---|
| | N = 1032 | | N = 176665 | |
| | N | % | N | % |
| 2012 | 193 | 11.0 | 28840 | 10.6 |
| 2013 | 136 | 7.7 | 25242 | 9.2 |
| 2014 | 149 | 8.5 | 25092 | 9.2 |
| 2015 | 127 | 7.2 | 24328 | 8.9 |
| 2016 | 120 | 6.8 | 23821 | 8.7 |
| **Parity** | | | | |
| First born | 557 | 31.7 | 93882 | 34.4 |
| Second born | 497 | 28.3 | 81326 | 29.8 |
| Third born | 334 | 19.0 | 51084 | 18,7 |
| Fourth born or later | 369 | 21.0 | 46717 | 17.1 |
| Missing | | | 99 | |
| **Plural delivery** | | | | |
| Yes | 61 | 3.5 | 7409 | 2.7 |
| No | 1696 | 96.5 | 265699 | 97.3 |
| **Preterm birth (gestational age <37 weeks)** | | | | |
| Yes | 261 | 14.9 | 25037 | 9.2 |
| No | 1488 | 85.1 | 247358 | 90.8 |
| Missing | - | | 713 | |
| Gestational age in weeks (SD) | 38.1 | 2.6 | 38.6 | 2.0 |
| **County size where child was born[a]** | | | | |
| Large central metro | 546 | 31.1 | 78540 | 28.8 |
| Large fringe metro | 135 | 7.7 | 25553 | 9.4 |
| Medium metro | 247 | 14.1 | 39540 | 14.5 |
| Small metro | 448 | 25.5 | 70671 | 25.9 |
| Micropolitan | 195 | 11.1 | 32149 | 11.8 |
| Non-core | 186 | 10.6 | 26655 | 9.8 |

SD: standard deviation; IQR: interquartile range.

[a] If childbirth county was missing maternal residential county was used.

Cells with sample sizes <10 are suppressed.

education) compared of those with genetic conditions (36.2%), cerebral palsy (24.7%), and autism (27.0%).

## Occurrence of pregnancy complications and other maternal outcomes

Gestational diabetes and gestational hypertension were slightly more common in women with IDD compared to women without IDD (Table 2). Women with IDD were more likely to be transferred to another medical facility during labor compared to women without IDD. Caesarean delivery was more common in women with IDD compared to women without IDD. Results were robust and did not meaningfully change when using two-claims for IDD, excluding births in 2007, 2013, and 2014, or restricting to just Medicaid covered deliveries (S3 and S4 Tables).

**Table 2. Occurrence and risk ratios of maternal pregnancy complications and maternal birth outcomes for all births comparing women with and without intellectual and developmental disabilities in Wisconsin Medicaid, 2007–2016.**

| | Pregnancies of women with Intellectual and developmental disabilities | Pregnancies of women without intellectual and developmental disabilities | Unadjusted risk ratio[a] | |
|---|---|---|---|---|
| | N = 1757 | N = 273108 | | |
| | % | % | RR | 95% CI |
| **Service use** | | | | |
| **Trimester prenatal care began** | | | | |
| 1 | 72.8 | 73.4 | 0.99 | 0.9, 1.0 |
| 2 | 21.4 | 21.6 | | |
| 3 | 4.2 | 4.1 | | |
| None | 1.6 | 0.8 | **1.83** | 1.2, 2.8 |
| Missing (N) | 54 | 8069 | | |
| **WIC[c] during pregnancy** | | | | |
| Yes | 79.6 | 66.6 | **1.20** | 1.1, 1.3 |
| No | 20.4 | 33.4 | | |
| Missing (N) | 25 | 3139 | | |
| **Pregnancy complications and maternal birth outcomes** | | | | |
| **Prepregnancy BMI[b]** | | | | |
| Underweight[c] | 4.6 | 3.1 | **1.67** | 1.2, 2.3 |
| Normal weight | 30.9 | 37.2 | | |
| Overweight | 24.5 | 26.0 | | |
| Obese[d] | 40.1 | 33.7 | **1.22** | 1.1, 1.3 |
| Missing (N) | 22 | 3363 | | |
| **Ever smoke during pregnancy** | | | | |
| Yes | 29.9 | 26.6 | **1.09** | 1.0, 1.2 |
| No | 70.1 | 73.4 | | |
| Missing (N) | 16 | 1255 | | |
| **Gestational diabetes** | | | | |
| Yes | 7.0 | 5.5 | **1.28** | 1.0, 1.6 |
| No | 93.0 | 94.5 | | |
| **Gestational hypertension** | | | | |
| Yes | 6.0 | 5.0 | **1.22** | 1.0, 1.5 |
| No | 94.0 | 95.0 | | |
| **Complications of delivery** | | | | |
| **Maternal transfer** | | | | |
| Yes | 1.5 | 0.8 | **1.86** | 1.2, 2.8 |
| No | 98.5 | 99.2 | | |
| Missing (N) | - | 313 | | |
| **Prolonged labor** | | | | |
| Yes | 1.1 | 1.1 | 0.95 | 0.6, 1.5 |

*(Continued)*

**Table 2.** (Continued)

| | Pregnancies of women with Intellectual and developmental disabilities | Pregnancies of women without intellectual and developmental disabilities | Unadjusted risk ratio[a] | |
|---|---|---|---|---|
| | N = 1757 | N = 273108 | | |
| | % | % | RR | 95% CI |
| No | 98.9 | 98.9 | | |
| Missing (N) | - | 300 | | |
| **Precipitous labor** | | | | |
| Yes | 4.8 | 4.2 | 1.17 | 0.9, 1.4 |
| No | 95.2 | 95.8 | | |
| Missing (N) | - | 300 | | |
| **Induced labor** | | | | |
| Yes | 28.4 | 27.5 | 1.02 | 0.9, 1.1 |
| No | 71.6 | 72.5 | | |
| Missing (N) | - | 225 | | |
| **Caesarean delivery** | | | | |
| Yes | 26.3 | 20.8 | **1.30** | 1.2, 1.4 |
| No | 73.7 | 79.2 | | |
| Missing (N) | 101 | 12135 | | |

CI: confidence interval; WIC: Special Supplemental Nutrition Program for Women Infants and Children; BMI: body mass index.

[a] Multilevel regression clustered by mother.

[b] Data only available for birth years 2011–2016.

[c] Comparison group is normal weight.

Bold indicates statistical significance at an alpha = 0.05 level.

Cells with values <10 are suppressed.

## Adjusted risk ratios comparing women with IDD to the Medicaid sample

Adjustment attenuated the association between WIC (aRR: 1.11; 95% CI: 1.0, 1.2) or pre-pregnancy obesity (aRR: 1.14; 95% CI: 1.0, 1.3) and IDD (Fig 1). The associations between gestational diabetes (aRR: 1.37, 95% CI:1.1, 1.7) or gestational hypertension (aRR: 1.30, 1.0, 1.7) and IDD were strengthened after adjustment. There was no meaningful change in estimate for caesarean delivery after adjustment (aRR: 1.33, 1.2, 1.5). In a post-hoc analysis, we restricted to nulliparous women with singleton, vertex, term births and found women with IDD had 1.20 times the risk of caesarean delivery compared to women without IDD (95% CI: 0.9, 1.5). Stratifying by each of the restrictions, caesarean delivery risk did not differ for women with and without IDD by levels of term birth, nulliparous birth, or singleton birth. Women with IDD had significantly higher risk of caesarean delivery compared to women without IDD if the infant was not breach and if the mother had no previous caesarean deliveries (S5 Table).

## Results by IDD-type

Women with intellectual disability were less likely to receive prenatal care in the first trimester compared to women without any IDD (Table 3). Women with a genetic condition had a higher risk of gestational diabetes and women with intellectual disability had a higher risk of gestational hypertension compared to women without IDD. Women with intellectual

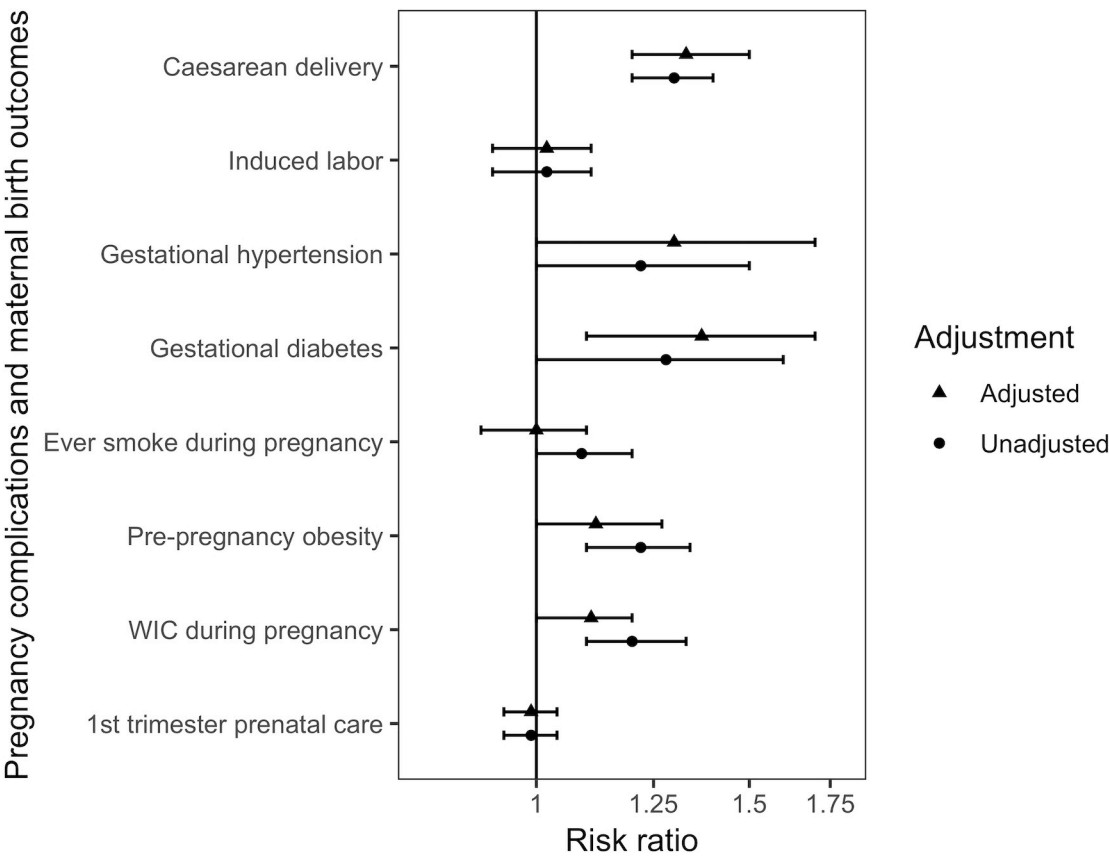

**Fig 1. Unadjusted and adjusted risk ratios for prevalent pregnancy complications and maternal birth outcomes for all births comparing women with and without intellectual and developmental disabilities in Wisconsin Medicaid, 2007–2016.** Multilevel regression clustered by mother. Adjusted for maternal age, race, ethnicity, foreign born mother, geographic classification of birth county size, parity, marriage, year. Obesity and WIC Data only available for birth years 2011–2016.

disability, genetic conditions, and cerebral were at higher risk of caesarean delivery compared to women without IDD.

## Interaction by race

When assessing differences by race and IDD (Table 4), results indicated that Black women with or without IDD were less likely than white women with or without IDD to receive first trimester prenatal care, ever smoke during pregnancy, or have gestational diabetes. Black women were more likely to be on WIC or have gestational hypertension, with additional excess risk for women with IDD; the race by IDD interaction term was statistically significant for gestational hypertension (P for interaction = 0.07) and WIC utilization (P = 0.02) indicating Black women with IDD were at higher risk than would expected based on risk in Black women and women with IDD independently.

## Discussion

Women with IDD face health inequity and disparity that may impact pregnancy outcomes. In this study, we replicated past findings that identify increased risk of poor maternal birth outcomes for pregnant women with IDD in the Wisconsin Medicaid system. We expanded upon previous findings, assessing complications and maternal birth outcomes by IDD-type and exploring statistical interaction between race and IDD.

**Table 3. Occurrence and unadjusted risk ratios of pregnancy complications and maternal birth outcomes for all births to women with intellectual and developmental disability types compared to women without intellectual and developmental disabilities, 2007–2016.**

| | Live births to women by intellectual and developmental disability type | | | | | | | | | | | |
|---|---|---|---|---|---|---|---|---|---|---|---|---|
| | Intellectual disability | | | Genetic condition | | | Cerebral palsy | | | Autism | | |
| | N = 552 | | | N = 777 | | | N = 279 | | | N = 156 | | |
| | % | RR[a] | 95% CI | % | RR[a] | 95% CI | % | RR[a] | 95% CI | % | RR[a] | 95% CI |
| **Service use** | | | | | | | | | | | | |
| **Trimester prenatal care began** | | | | | | | | | | | | |
| 1 | 65.4 | **0.89** | 0.8, 1.0 | 76.7 | 1.04 | 0.9,1.1 | 77.8 | 1.06 | 1.0, 1.1 | 74.3 | 1.02 | 0.9, 1.1 |
| 2 | 25.7 | | | 20.7 | | | 17.2 | | | 14.5 | | |
| 3 | 6.0 | | | 1.9 | | | 4.0 | | | 9.2 | | |
| None | 3.0 | | | - | | | | | | | | |
| Missing (N) | 15 | | | 27 | | | | | | | | |
| **WIC[b] during pregnancy** | | | | | | | | | | | | |
| Yes | 86.2 | **1.31** | 1.2, 1.4 | 74.2 | **1.11** | 1.0, 1.2 | 79,7 | **1.25** | 1.1, 1.4 | 79.8 | **1.20** | 1.1, 1.3 |
| No | 13.4 | | | 25.8 | | | 16.1 | | | 20.2 | | |
| **Pregnancy Complications and maternal birth outcomes** | | | | | | | | | | | | |
| **Prepregnancy BMI[b]** | | | | | | | | | | | | |
| Underweight | 6.7 | | | 4.1 | | | | | | | | |
| Normal weight | 28.5 | | | 27.9 | | | 40.6 | | | 37.4 | | |
| Overweight | 22,7 | | | 27.4 | | | 24.6 | | | 28.6 | | |
| Obese[c] | 42.7 | **1.38** | 1.2, 1.6 | 40.7 | **1.22** | 1.1, 1.4 | 34.8 | 0.98 | 0.7, 1.3 | 34.1 | 1.06 | 0.8, 1.4 |
| **Ever smoke during pregnancy** | | | | | | | | | | | | |
| Yes | 31.6 | 1.11 | 0.9. 1.3 | 26.9 | 1.00 | 0.9, 1.2 | 28.8 | 1.03 | 0.8, 1.3 | 35.3 | 1.27 | 0.9, 1.6 |
| No | 68.5 | | | 73.1 | | | 71.2 | | | 64.7 | | |
| **Gestational diabetes** | | | | | | | | | | | | |
| Yes | 6.7 | 1.20 | 0.8, 1.7 | 8.4 | **1.54** | 1.2, 2.0 | 5.4 | 0.96 | 0.5, 1.8 | 6.4 | 1.22 | 0.6, 2.3 |
| No | 93.3 | | | 91.6 | | | 94.6 | | | 93.6 | | |
| **Gestational hypertension** | | | | | | | | | | | | |
| Yes | 8.7 | **1.71** | 1.3, 2.3 | 4.6 | 1.00 | 0.7, 1.3 | 3.9 | 0.73 | 0.4, 1.4 | 7.7 | 1.60 | 0.9, 2.8 |
| No | 91.3 | | | 95.4 | | | 96.1 | | | 92.3 | | |
| **Complications of delivery** | | | | | | | | | | | | |
| **Induced labor** | | | | | | | | | | | | |
| Yes | 26.7 | 0.96 | 0.8, 1.1 | 31.6 | **1.13** | 1.0, 1.3 | 21.9 | 0.81 | 0.6, 1.0 | 28.6 | 1.03 | 0.8, 1.4 |
| No | 73.3 | | | 68.4 | | | 78.1 | | | 71.4 | | |
| **Caesarean delivery** | | | | | | | | | | | | |
| Yes | 23.8 | **1.25** | 1.1, 1.5 | 29.7 | **1.29** | 1.1, 1.5 | 35.7 | **1.63** | 1.3, 2.0 | 24.3 | 1.16 | 0.8, 1.6 |
| No | 76.2 | | | 70.3 | | | 64.3 | | | 75.7 | | |
| Missing | 40 | | | 42 | | | 21 | | | | | |

CI: confidence interval; WIC: Special Supplemental Nutrition Program for Women Infants and Children; BMI: body mass index, RR: risk ratio.

[a] Multilevel regression clustered by mother comparing intellectual and developmental disability type to women without intellectual and developmental disability.

[b] Data only available for birth years 2011–2016.

[c] Comparison group is normal weight.

Bold indicates statistical significance at an alpha = 0.05 level.

Cells with values <10 are suppressed.

Our findings using birth records are in line with past research on occurrence of outcomes. Using data from the 2010 US Nationwide Inpatient Sample, Parish et al. [20] found that 7.0% of pregnant women with IDD had gestational diabetes and 49.0% had caesarean deliveries.

**Table 4. Occurrence and unadjusted risk ratios for pregnancy complications and maternal birth outcomes comparing births to white women with and without intellectual and developmental disabilities and black women with and without intellectual and developmental disabilities in Wisconsin Medicaid, 2007–2016.**

| | White | | | | Black | | | | | |
| | IDD | No-IDD | RR | | IDD | RR | | No-IDD | RR | |
| | N = 1220 | N = 196684 | IDD white vs. No-IDD white | | N = 394 | IDD black vs. No-IDD white | | N = 52414 | No-IDD black vs. No-IDD white | Interaction P value |
| | % | % | RR[a] | 95% CI | % | RR[a] | 95% CI | % | RR[a] | 95% CI | |
|---|---|---|---|---|---|---|---|---|---|---|---|
| **Service use** | | | | | | | | | | | |
| **Trimester prenatal care began** | | | | | | | | | | | |
| 1 | 77.7 | 76.7 | 1.01 | 0.9, 1.0 | 65.2 | **0.85** | 0.8, 0.9 | 66.5 | **0.87** | 0.8, 0.9 | 0.5 |
| 2 | 18.7 | 19.2 | | | 24.9 | | | 25.7 | | | |
| 3 | 3.6 | 3.5 | | | 5.1 | | | 5.8 | | | |
| None | | 0.6 | | | 4.8 | | | 2.0 | | | |
| Missing (N) | 41 | 4988 | | | - | | | 2458 | | | |
| **WIC[b] during pregnancy** | | | | | | | | | | | |
| Yes | 77.3 | 62.2 | **1.24** | 1.2, 1.3 | 82.6 | **1.34** | 1.2, 1.4 | 78.3 | **1.26** | 1.2, 1.3 | **<0.001** |
| No | 22.7 | 37.8 | | | 17.4 | | | 27.7 | | | |
| Missing (N) | 15 | 2164 | | | - | | | 608 | | | |
| **Pregnancy complications and maternal birth outcomes** | | | | | | | | | | | |
| **Prepregnancy BMI[b]** | | | | | | | | | | | |
| Underweight | 4.4 | 3.1 | | | 6.1 | | | 3.1 | | | |
| Normal weight | 30.1 | 37.9 | | | 31.1 | | | 32.9 | | | |
| Overweight | 23.8 | 25.9 | | | 25.5 | | | 25 | | | |
| Obese | 41.6 | 33.1 | **1.29** | 1.2, 1.4 | 37.2 | 1.16 | 0.9, 1.4 | 39 | **1.18** | 1.1, 1.2 | **0.02** |
| Missing (N) | 11 | 2927 | | | - | | | 666 | | | |
| **Ever smoke during pregnancy** | | | | | | | | | | | |
| Yes | 33.4 | 29.8 | 1.07 | 0.9, 1.2 | 22.6 | **0.78** | 0.6, 0.9 | 19.1 | **0.62** | 0.6, 0,7 | 0.2 |
| No | 66.6 | 70.2 | | | 77.4 | | | 80.9 | | | |
| Missing (N) | | 816 | | | - | | | 255 | | | |
| **Gestational diabetes** | | | | | | | | | | | |
| Yes | 7.1 | 5.5 | **1.30** | 1.0, 1.6 | 4.2 | 0.71 | 0.4, 1.3 | 4.1 | **0.75** | 0.7, 0.8 | 0.3 |
| No | 92.9 | 94.5 | | | 95.9 | | | 95.9 | | | |
| **Gestational hypertension** | | | | | | | | | | | |
| Yes | 4.5 | 4.7 | 1.02 | 0.8, 1.3 | 10.5 | **2.24** | 1.5, 3.1 | 7.2 | **1.51** | 1.4, 1.6 | **0.07** |
| No | 95.5 | 95.3 | | | 89.5 | | | 92.9 | | | |
| **Complications of delivery** | | | | | | | | | | | |
| **Induced delivery** | | | | | | | | | | | |
| Yes | 29.6 | 28.4 | 1.03 | 0.9, 1.1 | 27.2 | 0.96 | 0.8, 1.1 | 27.2 | 0.97 | 0.9, 1.0 | 0.8 |
| No | 70.4 | 71.6 | | | 72.3 | | | 72.8 | | | |
| Missing (N) | | 140 | | | - | | | 58 | | | |
| **Caesarean delivery** | | | | | | | | | | | |
| Yes | 26.9 | 21.5 | **1.30** | 1.2, 1.5 | 25.1 | **1.32** | 1.1, 1.6 | 20.2 | 0.96 | 0.9, 1.0 | 0.6 |
| No | 73.1 | 78.5 | | | 74.9 | | | 79.8 | | | |
| Missing | 65 | 9192 | | | 11 | | | 2318 | | | |

IDD: intellectual and developmental disabilities; CI: confidence interval; WIC: Special Supplemental Nutrition Program for Women Infants and Children; BMI: body mass index; RR: risk ratio.

[a] Multilevel regression clustered by mother, with race term and race by intellectual and developmental disability interaction term; referent group is white women without IDD.

[b] Data only available for birth years 2011–2016.

[c] Comparison group is normal weight.

Bold indicates statistical significance at an alpha = 0.1 level.

Cells with values <10 are suppressed.

Occurrence of caesarean delivery was higher than in our sample which may be an effect of temporal trends related to decreasing national rates of caesarean delivery [24], or may be a result of the lower rates of caesarean delivery in Wisconsin [25]. Mitra et al. [26, 27] used data from the Massachusetts Pregnancy to Early Life Longitudinal Data system and found similar occurrence of gestational diabetes (5.8%), gestational hypertension (5.8%), and caesarean delivery (36.0%) in women with IDD. Our results were also in line with Darney et al's [28] work in a retrospective cohort of all live births in California from 2000–2010 (gestational diabetes: 8.1%; caesarean delivery: 38.0%). In the Ontario health system from 2002–2010, Brown et al. [29] found lower rates of gestational diabetes (3.2%) and gestational hypertension (1.2%) compared to our findings with similar rates of caesarean delivery (27.0%). With these consistent results, obstetricians should be aware of the high risk of maternal morbidity when treating pregnant women with IDD and researchers need to continue to investigate mechanistic pathways and IDD-specific interventions and preventive programs.

When comparing women with IDD to a non-IDD sample, our estimates align with the recent meta-analysis by Tarasoff et al. [11] For gestational diabetes, our uRR of 1.28 was larger than the meta-analytic unadjusted pooled odds ratio of 1.10 (assuming odds ratio approaches RR) yet was within the range of other studies (odds ratio range: 0.77–1.71). Our uRR for gestational hypertension (1.22) was smaller than the meta-analytic estimate (odds ratio 1.77) but within previous studies' range (0.63–2.49). Lastly, Tarasoff et al. [11] estimated unadjusted and adjusted odds ratios for caesarean delivery finding an unadjusted and adjusted pooled odds ratio of 1.29 and 1.46 respectively. The pooled odds ratio for caesarean delivery (1.46) was higher than our estimate (aRR 1.29), possibly due to the odds ratio overestimating the risk ratio given the high prevalence of caesarean delivery. We expected variation compared to past work because of differing demographics and methods, including the relatively low rates of caesarean delivery in Wisconsin [30]; however, the direction and magnitude of effects are similar to past work and provide further evidence of increased risk for maternal pregnancy complications for women with IDD. Promisingly, we found no difference in uptake of prenatal care services, which is in contrast to the disparity in reproductive health services often seen for women with IDD [9]. In this analysis, we identified outcomes a priori based on existing literature; in the future we aim to link more data and use Big Data methodology, such as machine learning, to explore novel risk factors and interactions for pregnancy outcomes for women with IDD.

We saw minimal changes when statistically adjusting for demographic differences. Our use of a Medicaid sample reference group was selected to reduce and account for much of the confounding due to socio-economic differences seen in previous studies. With additional data, we will have the statistical power to adjust our IDD-subtype analyses and better explore specific mediators and causal pathways from IDD to outcome.

With the heterogeneity of IDD, it was important to assess outcomes in by IDD type. Women with intellectual disability were less likely to receive prenatal care in the first trimester compared to women without IDD. The lack of adequate reproductive and sexual health care [9] may delay recognition of pregnancy and start of services. In addition, women with intellectual disability may face increased barriers when navigating the health care service system [31], especially for pregnancy care [9]. The increased risk for caesarean delivery for women with cerebral palsy may be due to chronic pain, pelvic abnormalities, spasticity, and other physical disabilities [12]. Because of the approximate 4:1 male to female sex ratio in autism [32], and temporal cohort effects [3], we saw relatively fewer births to women on the autism spectrum. While we present what we believe to be the first population-based estimates of pregnancy outcomes specific to autistic women, these data will be bolstered as more women on the spectrum age into adulthood and start families.

We found some differences in risk of pregnancy complications between Black and white women with IDD, with Black women having increased risks of obesity and gestational

hypertension and white women having increased risks of smoking during pregnancy and gestational diabetes. It is possible that pre-pregnancy diabetes and hypertension may influence this finding and will need further exploration. For other outcomes, such as caesarean delivery and prenatal care, we found little evidence of excess risk for being both black and having IDD. Based on our results in pregnancies covered by Wisconsin Medicaid, much of the racial disparity in complications experienced by white and Black women with IDD is not different from racial disparity from white and Black women without IDD.

Determination of both maternal pregnancy outcomes and IDD could be impacted by misclassification. Our reliance on birth records to quantify outcomes may lead to an underestimation of morbidity, especially for gestational hypertension and diabetes, and maternal smoking [33, 34]. Ultimately, our estimates were consistent with past studies that used other data sources for maternal outcomes [26, 27]. Because of the demographic similarities between the IDD and non-IDD group in our sample and the minimal effect of statistical adjustment, differential misclassification biasing RRs was unlikely. Our findings need to be replicated with other data sources such as maternal self-report, electronic health records, and claims. With additional data sources we can better describe and understand prenatal care and perinatal management in women with IDD. IDD was identified using up to one year of maternal Medicaid claims as available in BD4LK; we are not able to determine IDD status in women on Medicaid who did not have claims for IDD. It would have been preferable to have a longer period to assess claims or additional data sources. However, our results were consistent when using more restrictive IDD claim criteria. Our sample included women who received Medicaid and does not represent the full population of women with IDD. We cannot make inferences on women with IDD on private insurance who may have different socioeconomic profiles. IDD often co-occur [35], which was not evident in our claims; our categorization by IDD-type may not have captured the true overlap of some conditions and additional data are needed to better capture IDD phenotype. Our results are conditioned on live birth and do not account for pregnancy losses. Results may not be generalizable to other state Medicaid systems due to the demographic distribution and state-specific policies of Wisconsin.

## Conclusions

Pregnant women with IDD and a live birth in Wisconsin Medicaid were at greater risk of gestational diabetes, gestational hypertension, and caesarean delivery compared to pregnant women without IDD. Our findings are in line with past studies and highlight the importance of proper accounting for socioeconomic status and exploring IDD-type and race. Results support the need for increased research and attention to maternal pregnancy complications and adverse birth outcomes for women with IDD. Further work is needed to deduce biologic and social mechanisms for the presentation of complications.

## Supporting information

**S1 Table. International classification of disease 9 and 10 codes used to identify intellectual and developmental disabilities.**
(DOCX)

**S2 Table. Demographic characteristics of mothers with a live Medicaid covered birth in Wisconsin, 2007–2016; by Intellectual and developmental disability type.**
(DOCX)

**S3 Table. Demographic characteristics of mothers with live birth in Wisconsin 2007–2016 comparing intellectual and developmental disability identification criteria in Medicaid**

**claims one-year pre pregnancy.**
(DOCX)

**S4 Table. Occurrence and risk ratios of maternal pregnancy complications and adverse outcomes for all births to mothers with intellectual and developmental disabilities compared to the full Wisconsin Medicaid sample of mothers, 2007–2016 with sensitivity analyses for IDD claim count and excluding years.**
(DOCX)

**S5 Table. Analysis of caesarean delivery in women with and without intellectual and developmental disabilities in Wisconsin Medicaid, 2007–2016.**
(DOCX)

**S1 File. Birth outcomes affecting infants of mothers with intellectual and developmental disabilities.**
(PDF)

## Acknowledgments

Due to our data use agreement with the Wisconsin Department of Health Services we are unable to share data. Please contact the corresponding author for information on any code.

The authors of this article are solely responsible for the content therein. The authors would like to thank the Department of Health Services, for the use of data for this analysis, but these agencies do not certify the accuracy of the analyses presented.

We thank Steven T. Cook, and Kristen Voskuil, for data access and programming assistance. We also thank the Wisconsin Department of Children and Families, Department of Health Services for the use of their data.

## Author Contributions

**Conceptualization:** Eric Rubenstein, Deborah B. Ehrenthal, Lauren Bishop.

**Data curation:** Eric Rubenstein, Deborah B. Ehrenthal, David C. Mallinson, Hsiang-Huo Kuo.

**Formal analysis:** Eric Rubenstein.

**Funding acquisition:** Eric Rubenstein, Deborah B. Ehrenthal, Lauren Bishop, Maureen Durkin.

**Investigation:** Eric Rubenstein.

**Methodology:** Eric Rubenstein, Deborah B. Ehrenthal, David C. Mallinson, Hsiang-Huo Kuo, Maureen Durkin.

**Project administration:** Eric Rubenstein, Deborah B. Ehrenthal.

**Resources:** Maureen Durkin.

**Software:** Deborah B. Ehrenthal.

**Supervision:** Deborah B. Ehrenthal, Hsiang-Huo Kuo, Maureen Durkin.

**Validation:** Eric Rubenstein, David C. Mallinson, Hsiang-Huo Kuo.

**Visualization:** Eric Rubenstein.

**Writing – original draft:** Eric Rubenstein.

**Writing – review & editing:** Eric Rubenstein, Deborah B. Ehrenthal, David C. Mallinson, Lauren Bishop, Hsiang-Huo Kuo, Maureen Durkin.

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
