## [Decision Letter · Decision Letter 0]

17 Aug 2020

PONE-D-20-17994

Pregnancy complications and maternal birth outcomes in women with intellectual and developmental disabilities in Wisconsin Medicaid

PLOS ONE

Dear Dr. Rubenstein,

Thank you for submitting your manuscript to PLOS ONE. After careful consideration, we feel that it has merit but does not fully meet PLOS ONE’s publication criteria as it currently stands. Therefore, we invite you to submit a revised version of the manuscript that addresses the points raised during the review process.

ACADEMIC EDITOR: Very interesting article about an important subject with a big sample size. The article needs a major revision and the authors should add many obstetrical information about these specific pregnant women to interest the reader.

We look forward to receiving your revised manuscript.

Kind regards,

Guillaume Ducarme, MD, MSc, PhD

Academic Editor

PLOS ONE

Journal Requirements:

2. In ethics statement in the manuscript and in the online submission form, please provide additional information about the database used in your retrospective study. Specifically, please ensure that you have discussed whether all data were fully anonymized before you accessed them and/or whether the IRB or ethics committee waived the requirement for informed consent. If patients provided informed written consent to have their data used in research, please include this information.

3.We note that you have indicated that data from this study are available upon request. PLOS only allows data to be available upon request if there are legal or ethical restrictions on sharing data publicly. For information on unacceptable data access restrictions, please see http://journals.plos.org/plosone/s/data-availability#loc-unacceptable-data-access-restrictions.

Reviewers' comments:

Reviewer's Responses to Questions

**Comments to the Author**

1. Is the manuscript technically sound, and do the data support the conclusions?

Reviewer #1: Yes

Reviewer #2: Yes

2. Has the statistical analysis been performed appropriately and rigorously? 

Reviewer #1: Yes

Reviewer #2: Yes

3. Have the authors made all data underlying the findings in their manuscript fully available?

Reviewer #1: Yes

Reviewer #2: Yes

4. Is the manuscript presented in an intelligible fashion and written in standard English?

Reviewer #1: Yes

Reviewer #2: Yes

5. Review Comments to the Author

Reviewer #1: REVIEW: PONE-D-20-17994

Thanks to the editorial board to give me the opportunity to review this interesting works. The resuts of this study confirm well-known issues for women with Intellectual and Developmental Disabilities, nevertheless with a “big data” approach their provide us a confirmation of obstetrical outcomes in this population.

Minors Comments:

Methods

1- Could authors clarify the datas mining in this “Big data” study step by step. This could be helpful for further studies in the same methodology and for external validity / reproducibility of this study.

2- Authors assess obstetrical outcomes and c-section more particularly. This outcome is very difficult to assess because of numbers of biais leading to c-section. Could you precise, in the according to the numbers of patient and variables why a propensity score was not choice before adjustment ? This method is more and more recommended in big data analysis currently.

Results

1- In this big data analysis we can be surpize by the little numbers of missing values, How can authors explain that. Please discuss this in discussion.

2- Authors compared population characteristics in the table 1 without precision on the significance of comparisons done. Please test and add p-value in table 1, and report in the section results.

Discussion:

1- For non American readers I think it’s could be useful to precise what is medicare “a government insurance program for persons of all ages whose income and resources are insufficient to pay for health care”, in order to better precise the characteristic of this sample. Could the results observed in this singular sample of population be extrapolate in general population according to this insurance selection ? Please debates this.

2- The discussion of the results is welle done wizth an epidemiological point of view. An obstetrical vision in the discussion could be useful, because obstetricians could be interest by the result of this study, So discussion about indication of c-section in this context is necessary. Indeed indication of c-section in the context of IDD could be done for obstetrical reason but also for organization reason, or obstetrician convenience considering the difficulties to manage such patient in labor ward for example.

Reviewer #2: The authors reported perinatal outcome of pregnancies in women with intellectual and developmental disabilities (IDD). Data were extracted from the Big Data for Little Kids project in Wisconsin Medicaid from 2007-2016. The authors showed that women with IDD have increased risk of pregnancy complications and adverse outcomes in Wisconsin Medicaid (greater risk for gestational, gestational hypertension, and caesarean delivery).

The subject is very interesting and the manuscript is very well-written. The sample size is very important (1032 women with IDD compared to 176665 women without IDD) and results seem robust to adjustment.

But, some problems decrease the interest and the authors should make modifications and add some major information about obstetric and perinatal management of these women.

No information about the gestational age at birth.

“Delivery complications”: Did the authors mean postpartum hemorrhage? Severe perineal tears? Operative vaginal delivery? Episiotomy?.... Many perinatal information lack in the text.

What is “precipitous labor”? Did they mean preterm labor?

Cesarean section: the authors should add some major information about the indications (failed labor progression, fetal distress…).

Same comment about the women who required nduced labor: why? Indications? Techniques? Results?

6. PLOS authors have the option to publish the peer review history of their article (what does this mean?). If published, this will include your full peer review and any attached files.

Reviewer #1: **Yes: **David Desseauve

Reviewer #2: **Yes: **Guillaume DUCARME

---

## [Author Response · Author response to Decision Letter 0]

16 Sep 2020

Maternal IDD outcomes response to reviewers 

We would like to thank the editors and reviewers for the positive and constructive review for our paper assessing maternal outcomes for women with intellectual and developmental disabilities. In addressing your concerns, we believe that we have a stronger, more robust paper. Additionally, we have taken the steps to meet Journal requirements. We have responded to comments and provide the page and line numbers for changes. In the text, changes are tracked

 We have made changes to meet the style requirements of PLOS ONE, including bolding the abstract

2. In ethics statement in the manuscript and in the online submission form, please provide additional information about the database used in your retrospective study. Specifically, please ensure that you have discussed whether all data were fully anonymized before you accessed them and/or whether the IRB or ethics committee waived the requirement for informed consent. If patients provided informed written consent to have their data used in research, please include this information.

We have added information on the waived consent and anonymization on page 7 line 20. 

3.We note that you have indicated that data from this study are available upon request. PLOS only allows data to be available upon request if there are legal or ethical restrictions on sharing data publicly. For information on unacceptable data access restrictions, please see http://journals.plos.org/plosone/s/data-availability#loc-unacceptable-data-access-restrictions.

We have added the following paragraph to the cover letter:

The data from this secondary data analysis are from the Big Data for Little Kids projects, which access proprietary Wisconsin state administrative data. As part of the larger study, the data manager has access to protected health information. While this study did not have access to those data, there is not yet a mechanism or agreement to share de-identified data with people outside the research team. Further, the larger projects Data Use Agreement with the Wisconsin Department of Health Services explicitly states no data can be shared publicly. The body governing the data is the University of Wisconsin Institute for Research on Poverty and for further inquiries you can contact Steve Cook (steven.cook@wisc.edu), IRP data sharing coordinator. 

Please see above explanation as to why we cannot share anonymized data. 

Reviewer #1: REVIEW: PONE-D-20-17994

Thanks to the editorial board to give me the opportunity to review this interesting works. The resuts of this study confirm well-known issues for women with Intellectual and Developmental Disabilities, nevertheless with a “big data” approach their provide us a confirmation of obstetrical outcomes in this population.

We thank the reviewer for their time and effort in providing us feedback. While there is existing literature on this topic, we believe that more is needed, and our study adds data from a different health system and geographic area. 

Minors Comments:

Methods

1- Could authors clarify the datas mining in this “Big data” study step by step. This could be helpful for further studies in the same methodology and for external validity / reproducibility of this study.

This study did not use any typical ‘data mining’ techniques. We developed an algorithm to identify IDD based on previous work and then examined a priori chosen outcomes. We have added that explanation on page 6 line 10. We have added a future direction of using big data techniques, e.g. machine learning, in the discussion (page 19 line 16).

2- Authors assess obstetrical outcomes and c-section more particularly. This outcome is very difficult to assess because of numbers of biais leading to c-section. Could you precise, in the according to the numbers of patient and variables why a propensity score was not choice before adjustment ? This method is more and more recommended in big data analysis currently.

While there may be differences in who gets C-sections, the goal of our study was to describe occurrence in women with IDD and compare to women without IDD. We believe that C-section is accurately defined in birth records (see Northam and Knapp 2006 doi: 10.1111/j.1552-6909.2006.00016.x) but acknowledge that some outcomes are not as reliable in the birth record. 

As for propensity scores, we believe that with our dataset standard statistical adjustment is a strong approach to address confounding and allows us to use the full Medicaid sample. Further, we saw little change with statistical adjustment using standard regression and would expect to see minor differences using a different method like propensity scores. 

Results

1- In this big data analysis we can be surpize by the little numbers of missing values, How can authors explain that. Please discuss this in discussion.

We have added to the discussion on page 21 line 13 discussing how we are missing data on women with IDD who do not have claims for IDD. That would not appear in the tables but is still missing. The birth record data has very little missingness, which is a credit to the reporting system. 

2- Authors compared population characteristics in the table 1 without precision on the significance of comparisons done. Please test and add p-value in table 1, and report in the section results.

We do not believe that testing for P values in table 1 is appropriate here. Since P values are an effect of sample size, we may see differences that are not clinically meaningful. Additionally, we used a conceptual framework to determine what to adjust for rather than statistical tests, so the P values would not inform our research. See the STROBE guideline for reporting observational studies for our reasoning in not including P values in table 1. (doi:10.1371/journal.pmed. 0040297, page 1643). 

Discussion:

1- For non American readers I think it’s could be useful to precise what is medicare “a government insurance program for persons of all ages whose income and resources are insufficient to pay for health care”, in order to better precise the characteristic of this sample. Could the results observed in this singular sample of population be extrapolate in general population according to this insurance selection ? Please debates this.

That is a terrific point. We have added a description of Medicaid in the introduction (page 3 line 24). We have also added to the discussion about how results may not be representative of a non-Medicaid sample (page 21 line 6). 

2- The discussion of the results is welle done wizth an epidemiological point of view. An obstetrical vision in the discussion could be useful, because obstetricians could be interest by the result of this study, So discussion about indication of c-section in this context is necessary. Indeed indication of c-section in the context of IDD could be done for obstetrical reason but also for organization reason, or obstetrician convenience considering the difficulties to manage such patient in labor ward for example.

We attempted to further examine c-section by doing a post-hoc analysis restricting to singleton, term, vertex birth. While not exactly the obstetrical reasons you mention, it does give us some insight into why this happens. We have added more information and supplement with this analysis (supplement 5, page 12, line 20).

We have added a sentence on the importance of the results to obstetricians (page 18 line 20). We do not want to overstate the implications or the actions to be taken from this study, but we acknowledge that with such high rates, obgyn’s should be aware of the increased risk of poor outcomes in women with IDD. 

Reviewer #2: The authors reported perinatal outcome of pregnancies in women with intellectual and developmental disabilities (IDD). Data were extracted from the Big Data for Little Kids project in Wisconsin Medicaid from 2007-2016. The authors showed that women with IDD have increased risk of pregnancy complications and adverse outcomes in Wisconsin Medicaid (greater risk for gestational, gestational hypertension, and caesarean delivery).

The subject is very interesting and the manuscript is very well-written. The sample size is very important (1032 women with IDD compared to 176665 women without IDD) and results seem robust to adjustment.

Thank you for your time and effort in reviewing our paper. We aimed to address your comments and concerns and believe that the manuscript is stronger having done so.

1. But, some problems decrease the interest and the authors should make modifications and add some major information about obstetric and perinatal management of these women.

No information about the gestational age at birth.

We have added gestational age to table 1. We have a separate analysis assessing infant outcomes which fully examines preterm birth, but we understand your point about it missing here. Our information on perinatal management is limited to prenatal care, which we present. As for perinatal management, we have added our lack of information as a limitation (page 21 line 6)

2.“Delivery complications”: Did the authors mean postpartum hemorrhage? Severe perineal tears? Operative vaginal delivery? Episiotomy?.... Many perinatal information lack in the text.

Thank you for pointing out that “delivery complications” is too ambiguous of a term. We have noted in the abstract that they are ‘common complications.’ Since we only had ~1000 women with IDD and we could not present outcomes with <10 occurrences (due to our data use agreement with the Department of Health Services), some of the rarer outcomes cannot be presented. We have added in our objective paragraph that we focused on common outcomes that we could assess in birth records (page 4 line 8) and made the same note when we talk about delivery complications (page 6 line 16)

3. What is “precipitous labor”? Did they mean preterm labor?

We have added the definition of precipitous and prolonged labor (page 6 line 16)

Preciptious labor is delivery after <3 hours of contractions, prolonged labor is >20 hours for first time mother or >14 hours for second time or greater mother. 

Cesarean section: the authors should add some major information about the indications (failed labor 4. progression, fetal distress…). Same comment about the women who required nduced labor: why? Indications? Techniques? Results?

Because of our reliance on birth records we were not able to pinpoint the indication of cesarean delivery or induction. For cesarean delivery, we did a post hoc analysis to see if effects were different between term, vertex, singleton births compared to other births. Risk was higher for preterm, breach, and multiple births. We have added tables of this analysis to the supplement 5 (page 12 line 20). In future work we hope to add electronic health records to get more detail on these conditions.

---

## [Decision Letter · Decision Letter 1]

13 Oct 2020

Pregnancy complications and maternal birth outcomes in women with intellectual and developmental disabilities in Wisconsin Medicaid

PONE-D-20-17994R1

Dear Dr. Rubenstein,

We’re pleased to inform you that your manuscript has been judged scientifically suitable for publication and will be formally accepted for publication once it meets all outstanding technical requirements.

Kind regards,

Guillaume Ducarme, MD, MSc, PhD

Academic Editor

PLOS ONE

Additional Editor Comments (optional):

Reviewers' comments:

Reviewer's Responses to Questions

**Comments to the Author**

1. If the authors have adequately addressed your comments raised in a previous round of review and you feel that this manuscript is now acceptable for publication, you may indicate that here to bypass the “Comments to the Author” section, enter your conflict of interest statement in the “Confidential to Editor” section, and submit your "Accept" recommendation.

Reviewer #1: All comments have been addressed

Reviewer #2: All comments have been addressed

2. Is the manuscript technically sound, and do the data support the conclusions?

Reviewer #1: Yes

Reviewer #2: Yes

3. Has the statistical analysis been performed appropriately and rigorously? 

Reviewer #1: Yes

Reviewer #2: Yes

4. Have the authors made all data underlying the findings in their manuscript fully available?

Reviewer #1: Yes

Reviewer #2: Yes

5. Is the manuscript presented in an intelligible fashion and written in standard English?

Reviewer #1: Yes

Reviewer #2: Yes

6. Review Comments to the Author

Reviewer #1: Thanks to the authors for the revision.

Major concern were solved by the authors and this paper could be accepted

Reviewer #2: All comments have been addressed. The revised manuscript is stronger after modifications.

The authors have strongly modified the manuscript to increase the interest for physicians, and, specifically, for obstetricians.

7. PLOS authors have the option to publish the peer review history of their article (what does this mean?). If published, this will include your full peer review and any attached files.

Reviewer #1: No

Reviewer #2: **Yes: **Guillaume DUCARME

---

## [Editor Report · Acceptance letter]

15 Oct 2020

PONE-D-20-17994R1 

Pregnancy complications and maternal birth outcomes in women with intellectual and developmental disabilities in Wisconsin Medicaid 

Dear Dr. Rubenstein:

I'm pleased to inform you that your manuscript has been deemed suitable for publication in PLOS ONE. Congratulations! Your manuscript is now with our production department. 

Kind regards, 

on behalf of

Dr. Guillaume Ducarme 

Academic Editor

PLOS ONE